# Structural variability and concerted motions of the T cell receptor – CD3 complex

Prithvi R Pandey[1], Bartosz Różycki[2], Reinhard Lipowsky[1], Thomas R Weikl[1]*

[1]Max Planck Institute of Colloids and Interfaces, Department of Theory and Bio-Systems, Potsdam, Germany; [2]Institute of Physics, Polish Academy of Sciences, Warsaw, Poland

**Abstract** We investigate the structural and orientational variability of the membrane-embedded T cell receptor (TCR) – CD3 complex in extensive atomistic molecular dynamics simulations based on the recent cryo-EM structure determined by Dong et al., 2019. We find that the TCR extracellular (EC) domain is highly variable in its orientation by attaining tilt angles relative to the membrane normal that range from 15° to 55°. The tilt angle of the TCR EC domain is both coupled to a rotation of the domain and to characteristic changes throughout the TCR – CD3 complex, in particular in the EC interactions of the Cβ FG loop of the TCR, as well as in the orientation of transmembrane helices. The concerted motions of the membrane-embedded TCR – CD3 complex revealed in our simulations provide atomistic insights on conformational changes of the complex in response to tilt-inducing forces on antigen-bound TCRs.

## Introduction

T cells recognize peptide antigens presented by major histocompatibility complexes (MHC) on apposing cell surfaces as a central step in the initiation of adaptive immune responses (*Rossjohn et al., 2015*; *Smith-Garvin et al., 2009*; *Dustin, 2014*; *Pettmann et al., 2018*; *Belardi et al., 2020*). The antigen recognition is performed by the T-cell receptor (TCR) complex, a complex of four dimeric transmembrane proteins. In this complex, the heterodimeric TCRαβ contains the binding site for recognizing peptide antigens, and the associated CD3εδ and CD3εγ hetero-dimers and the CD3ζζ homodimer contain the intracellular signaling motifs that transmit antigen binding to T cell activation (*Wucherpfennig et al., 2010*). While this stoichiometry of the complex has been known for nearly two decades (*Call et al., 2002*), the structure of the TCR – CD3 complex remained a puzzling problem (*Fernandes et al., 2012*; *Birnbaum et al., 2014*; *Natarajan et al., 2016*) that has only been recently solved by *Dong et al., 2019* with cryogenic electron microscopy (cryo-EM). To determine the structure, Dong et al. expressed all proteins of the complex in cultured cells, replaced the cell membrane around the assembled TCR – CD3 complex by the detergent digitonin, and stabilized the interactions between the extracellular (EC) domains of TCRαβ, CD3εδ and CD3εγ by chemical crosslinking. In the cryo-EM structure, the EC domains of CD3εδ and CD3εγ are both in contact with TCRαβ and with each other (see *Figure 1*), which explains the cooperative binding of CD3εδ and CD3εγ to TCRαβ observed in chain assembly (*Call et al., 2002*), mutational (*Kuhns and Davis, 2007*; *Kuhns and Davis, 2012*), and NMR experiments (*He et al., 2015*). As indicated by mutational experiments (*Kuhns and Davis, 2007*; *Kuhns and Davis, 2012*), the DE loop of the membrane-proximal constant domain Cα of TCRα is in contact with the CD3εδ EC domain, and the CC' loop of the constant domain Cβ of TCRβ is in contact with both the CD3εγ and CD3εδ EC domains in the assembled TCR – CD3 complex (see *Figure 1*). Outstanding questions concern the orientational variability of the TCRαβ EC domain relative to the membrane, in which the TCR – CD3

*For correspondence:
thomas.weikl@mpikg.mpg.de

**Competing interests:** The authors declare that no competing interests exist.

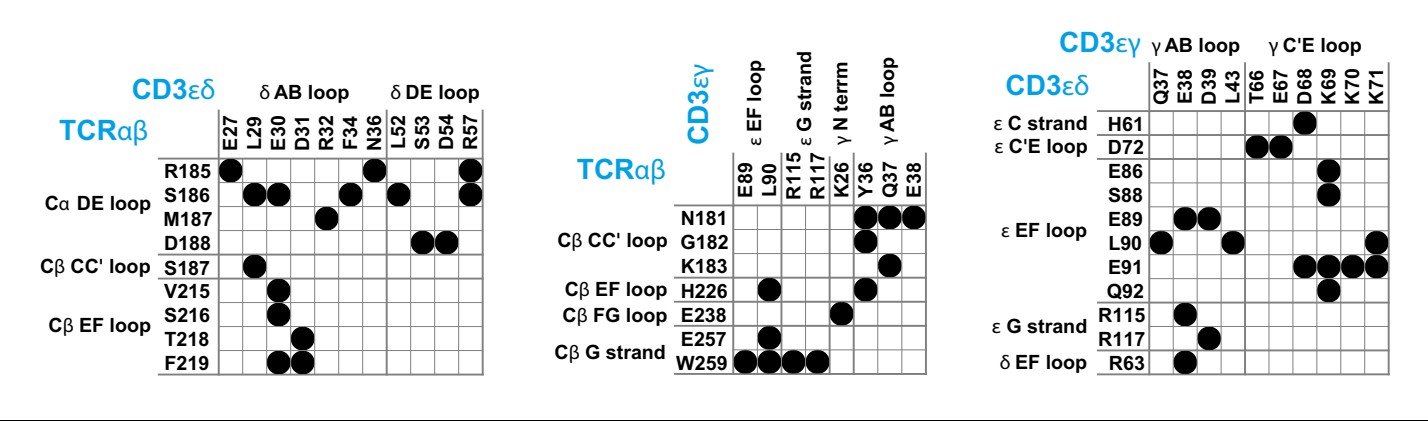

**Figure 1.** Maps of residue-residue contacts (black disks) between the EC domains of the protein dimers TCRαβ, CD3εδ, and CD3εγ in the cryo-EM structure of the T cell receptor – CD3 complex (*Dong et al., 2019*). Here, two residues are taken to be in contact if the minimum distance between non-hydrogen atoms of the residues is smaller than 0.45 nm. The loops and strands of the membrane-proximal constant domains Cα and Cβ of the proteins TCRα and TCRβ and of the EC domains of CD3ε, CD3γ, and CD3δ are labeled according to the standard convention for immunoglobulin-like domains (*Garcia et al., 1996*; *Wang et al., 1998*; *Sun et al., 2004*).

complex is embedded in its native environment, and the structural variability of the overall TCR – CD3 complex, which is constrained by the chemical crosslinking of the protein chains in the approach of *Dong et al., 2019*; *Reinherz, 2019*. The Cβ FG loop, for example, has been suggested to play a key role in T cell activation (*Kim et al., 2010*; *Touma et al., 2006*), but exhibits only rather limited contacts with CD3εγ in the cryo-EM structure (see *Figure 1*).

In this article, we investigate the structural and orientational variability of the membrane-embedded TCR – CD3 complex in extensive, atomistic molecular dynamics (MD) simulations with a cumulative simulation length of 120 μs. Compared to the cryo-EM structure, significantly more residues of TCRαβ are involved in EC domain contacts along our simulation trajectories, in particular in the Cβ FG loop, and notably also in the variable domain Vα of the TCRα chain. We find that the TCRαβ EC domain is rather variable in its orientation, with tilt angles relative to the membrane normal that range from 15° to 55°. The tilt of the TCRαβ EC domain is both coupled to a rotation of the domain and to characteristic changes in the overall structure of the TCR – CD3 complex. These structural changes include a clear decrease of contacts in the Cβ FG loop and an increase of contacts in the Vα domain with increasing tilt angle of the TCRαβ EC domain, as well as changes in the orientation of the transmembrane (TM) helices of the TCRα and CD3γ chain. The concerted motions of the membrane-embedded TCR – CD3 complex revealed in our simulations provide atomistic insights for force-based models of TCR signaling, which involve structural changes, in particular in the Cβ FG loop, that are induced by transversal, tilt-inducing forces on bound TCRs (*Brazin et al., 2015*; *Feng et al., 2018*).

## Results

Our computational analysis of the structural and orientational variability of the membrane-embedded TCR – CD3 complex is based on 120 atomistic, explicit-water MD simulation trajectories with a length of 1 μs and, thus, on simulation data with a total length of 120 μs. We have conducted these simulations with the Amber99SB-ILDN protein force field (*Lindorff-Larsen et al., 2010*) and the Amber Lipid14 membrane force field (*Dickson et al., 2014*) at a simulation temperature of 30°C on graphics processing units (GPUs). The simulation trajectories start from initial system conformations in which the cryo-EM structure of the TCR – CD3 protein complex is embedded in a membrane composed of 456 POPC lipids and 114 cholesterol molecules. We find that the orientational and conformational ensembles sampled by the 120 trajectories equilibrate within the first 0.5 μs of the simulation trajectories (see Materials and methods) and, therefore, focus on the second 0.5 μs of the MD simulation trajectories in our analysis.

Compared to the cryo-EM structure, a much larger set of residues is involved in contacts between the protein dimers of the TCR – CD3 complex in our MD simulations. *Figure 2* illustrates the time-

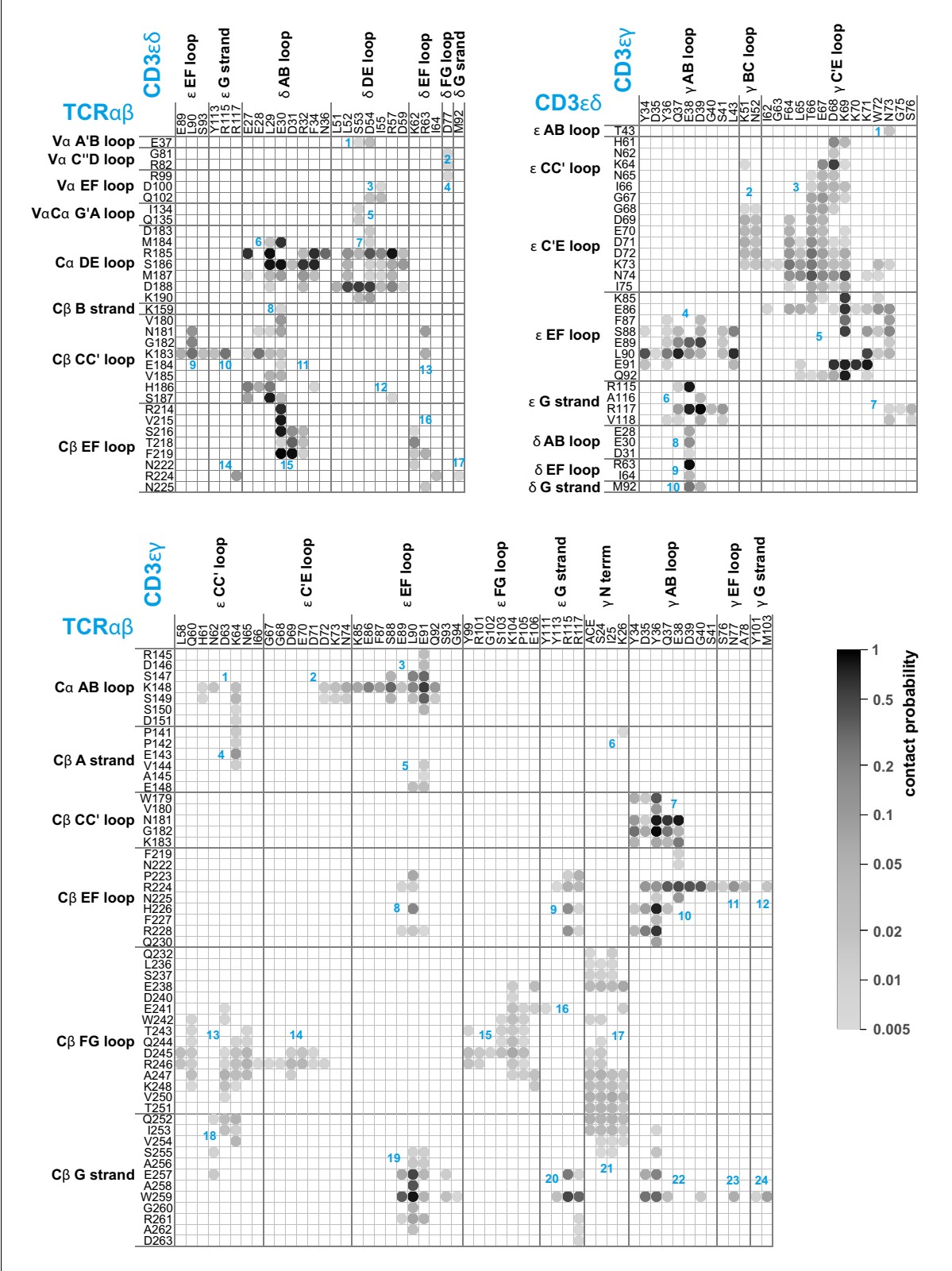

**Figure 2.** Averaged maps of contacts between the EC domains of TCRαβ, CD3εδ, and CD3εγ in the MD simulation trajectories. The shading of the contact disks indicates the contact probability, that is the fraction of simulation structures in which the contact is present. The contact analysis is based on 120 × 50 = 6000 structures extracted at intervals of 10 ns from the second halves of the 120 µs-long trajectories, which reflect an equilibrated ensemble of simulation conformations (see Materials and methods) and are available at the Edmond Open Research Data Repository (*Pandey and*

*Figure 2 continued on next page*

*Figure 2 continued*

*Weikl, 2021*). For clarity, only contacts with a contact probability larger than 0.5% are represented. As in *Figure 1*, two residues are taken to be in contact in a simulation structure if the minimum distance between non-hydrogen atoms of the residues is smaller than 0.45 nm. The contacts occur in clusters with numbers labeled in blue.

The online version of this article includes the following figure supplement(s) for figure 2:

**Figure supplement 1.** Spearman correlation cofficients of the contact clusters in the contact maps of *Figure 2*.

averaged contacts between residues of the TCRαβ, CD3εγ, and CD3εδ EC domains along the equilibrated second halves of the MD simulation trajectories. The fourth protein dimer in the complex, CD3ζζ, has no EC domain. The contact maps of *Figure 2* include all residue-residue contacts that occur in the simulations with a probability larger than 0.5%. The probabilities of the contacts are calculated from 6000 simulation structures extracted at intervals of 10 ns from the second 0.5 μs of the 120 simulations, and indicated in grayscale in *Figure 2*. As in the contact maps for the cryo-EM structure shown in *Figure 1*, two residues are taken to be in contact in a simulation structure if the minimum distance between non-hydrogen atoms of the residues is smaller than 0.45 nm. The contacts are grouped in clusters (blue numbers) that correspond to interactions between loops and strands of the EC domains, which are labeled according to the standard convention for immunoglobulin(Ig)-like domains (*Garcia et al., 1996*; *Wang et al., 1998*; *Sun et al., 2004*). The EC domains of the proteins TCRα and TCRβ consist of the membrane-proximal constant domains Cα and Cβ and the variable domains Vα and Vβ, which are all Ig-like domains, as are the EC domains of the proteins CD3ε, CD3γ, and CD3δ. In our MD simulations, significantly more loops and strands, and more residues of the protein dimers TCRαβ, CD3εγ, and CD3εδ participate in EC domain interactions, compared to the cryo-EM structure. The Cβ FG loop, for example, exhibits only a single contact with an N-terminal residue of CD3γ in the cryo-EM structure (see *Figure 1*). In our MD simulations, in contrast, the Cβ FG loop is involved in a large number of contacts with the N-terminus of CD3γ and with several loops and strand in the ε chain of the CD3εγ EC domain. Besides the Cα DE loop, Cβ CC' loop, Cβ EF loop, Cβ FG loop, and Cβ G strand with contacts in the cryo-EM structure, the MD contacts maps of *Figure 2* include also the Cα AB loop and the Cβ A and B strand in the constant domains of TCRαβ and, remarkably, the three loops A'B, C"D, and EF in the variable region Vα of TCRα. Residue-residue contacts between Vα and the δ chain of CD3εδ have probabilities smaller than 3%, but occur in 75 of the 120 trajectories and are, thus, a robust feature of our simulations. These contacts are grouped in four small, correlated contact clusters (see cluster-cluster correlation coefficients in *Figure 2—figure supplement 1*).

In our MD simulations, the TCRαβ EC domain is rather variable in its orientation relative to the membrane. The orientation can be quantified by two angles, a tilt angle and a rotation angle. To determine these angles, we choose two axes A and B in the TCRαβ EC domain: Axis A connects the centres of mass of Cαβ and Vαβ, where Cαβ is the dimer of the constant domains Cα and Cβ, and Vαβ is the dimer of the variable domains Vα and Vβ. Axis B connects the centres of mass of the variable domains Vα and Vβ. The tilt angle of the TCRαβ EC domain then is the angle between axis A and the membrane normal, and the rotation angle is the angle between axis B and the normal of the plane spanned by axis A and the membrane normal. The rotation angle describes the rotation of the TCRαβ EC domain around axis A (see *Figure 3(a) and (b)*). In our MD simulations, the tilt angle of TCRαβ EC domain roughly varies between 15° and 55°, while the rotation angle varies between 0° and 55°. A rotation angle of 0° indicates a TCRαβ EC domain orientation in which the centres of mass of the variable domains Vα and Vβ are equally close to the membrane, and the rotation angle is larger than 0° in conformations in which the variable domain Vα is closer to the membrane than the variable domain Vβ (see *Figure 3(a) and (b)*).

The two-dimensional probability distribution of the angles in *Figure 3(c)* indicates that the rotation of the TCRαβ EC domain is coupled to its tilt: For tilt angles between 15° and 35°, the rotation angle predominantly adopts values between about 5° and 25°. For a tilt angle of 40°, the most probable value of the rotation angle is about 32°, and further increases to 40° for a tilt angle of 50°. The coupling between the tilt and rotation of the TCRαβ EC domain is also illustrated in *Figure 3(a) and (b)*. In the structure of the membrane-embedded TCR – CD3 complex of *Figure 3(a)*, the TCRαβ EC domain has a tilt angle of 32.8° and a rotation angle of 12.8°. In the structure of *Figure 3(b)* with a larger tilt angle of 50.8°, the rotation angle of the TCRαβ EC domain is 42.9°. The rotation and tilt

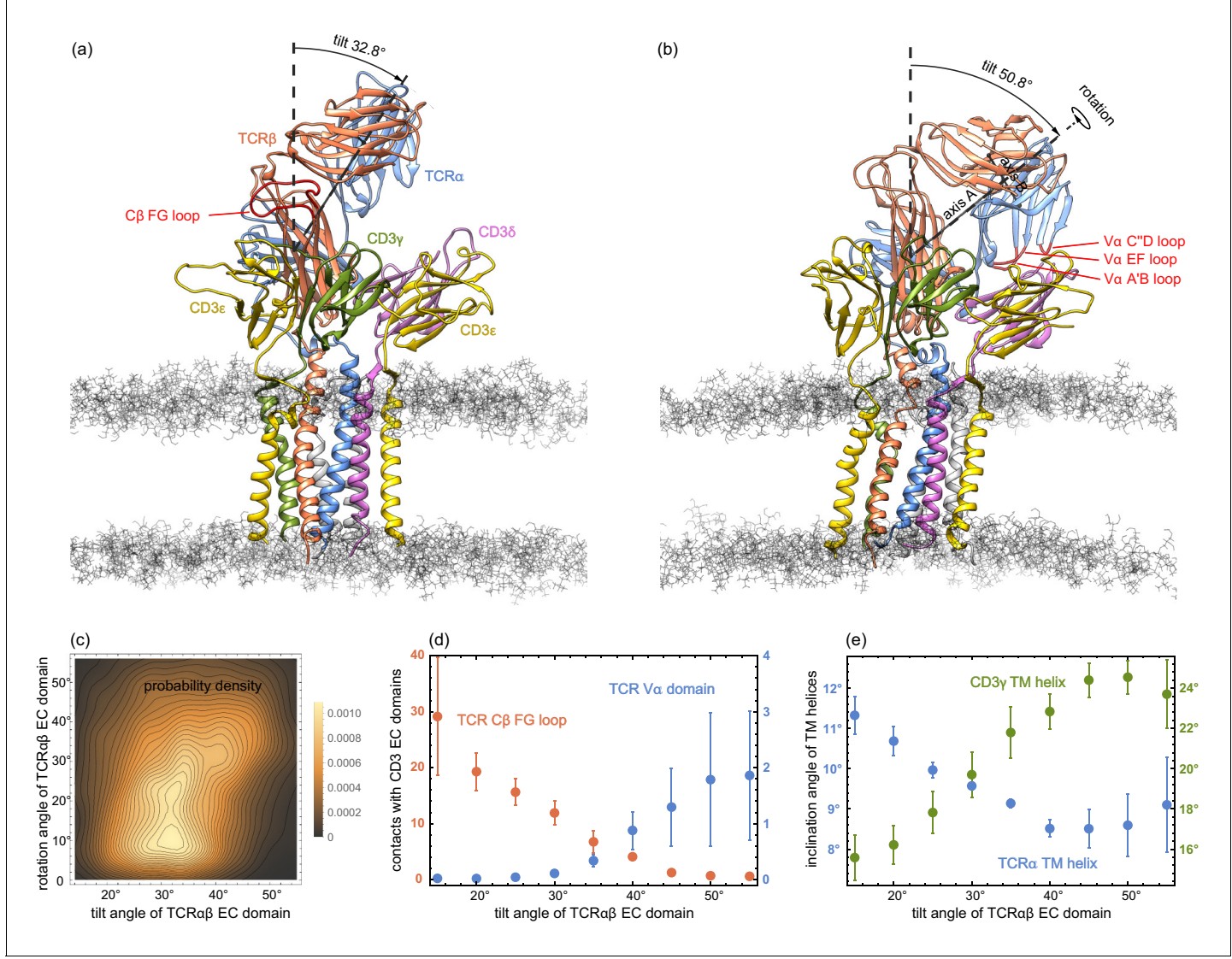

**Figure 3.** (a) and (b) MD conformations of the TCR – CD3 complex with different tilt angles of the TCRαβ ECdomain relative to the membrane normal. The rotation angles of the TCRαβ ECdomain are 12.8° and 42.9° in the conformations (a) and (b), respectively. (c) Two-dimensional probability density function for the tilt angle and rotation angle of the 6000 equilibrated MD conformations from the 120 trajectories. (d) Numbers of residue-residue contacts with CD3 EC domains for the Cβ FG loop and the Vα domain versus tilt angle. (e) Inclination angle of the TM helices in the TCRα and CD3γ chain relative to the membrane normal as a function of the tilt angle of the TCRαβ ECdomain. The errors in (d) and (e) have been estimated as error of the mean of averages obtained for five independent subsets of the MD conformations.

The online version of this article includes the following video and figure supplement(s) for figure 3:

**Figure supplement 1.** Numbers of residue-residue contacts of TCR Cα and Cβ loops and strands in interaction with CD3s versus tilt angle of the TCRαβ EC domain.

**Figure supplement 2.** Inclination angles of TM helices relative to the membrane normal as a function of the tilt angle of the TCRαβ ECdomain.

**Figure supplement 3.** (a) Force-free tilt-angle distribution of the TCRαβ EC domain obtained from our simulations (blue data points) and tilt-angle distributions under transversal forces $f = 2$ pN and 5 pN acting on the TCR-MHC complex, estimated from the force-free distribution; (b) local membrane thickness around the TM domain of the TCR – CD3 complex as a function of the tilt angle of the TCRαβ ECdomain.

**Figure supplement 4.** Tilt and rotation angle of the TCRαβ EC domain along the three exemplary trajectory segments shown in *Figure 3—videos 1, 2* and *3*.

**Figure 3—video 1.** Movie of trajectory segment 1.
https://elifesciences.org/articles/67195#fig3video1

**Figure 3—video 2.** Movie of trajectory segment 2.
https://elifesciences.org/articles/67195#fig3video2

**Figure 3—video 3.** Movie of trajectory segment 3.

*Figure 3 continued on next page*

*Figure 3 continued*

https://elifesciences.org/articles/67195#fig3video3

angle for the TCRαβ EC domain in the cryo-EM structure of the TCR – CD3 complex can be determined by aligning the TM domain of this structure to our simulation conformations. This structural alignment embeds and orients the cryo-EM structure in our simulated membranes. From TM domain alignment to the 120 final simulation conformations of our trajectories, we obtain the tilt angle $31.2\pm0.4°$ and the rotation angle $14.7\pm0.7°$ for the TCRαβ EC domain of the cryo-EM structure. The errors here have been estimated as the error of the mean of the 120 values obtained after structural alignment to these simulation conformations.

The tilt angle of the TCRαβ EC domain is associated with characteristic changes in the overall structure of the TCR – CD3 complex, in particular with changes in the number of residue-residue contacts of the Vα domain and of the Cβ FG loop (see *Figure 3(d)*) and in the orientations of the transmembrane (TM) helices of the TCRα and CD3γ chains (see *Figure 3(e)*). Residue-residue contacts of the A'B, C"D, and EF loops of the variable domain Vα with the protein CD3δ only occur for tilt angles of the TCRαβ EC domain larger than about 30° (see *Figure 3(d)*). The average number of these residues-residue contacts increases to values around two for tilt angles of 50° and larger. For the Cβ FG loop, in contrast, the average number of residues-residue contacts decreases from a value around 30 at the tilt angle 15° to values close 0 for tilt angles of 50° and larger. A decrease in the average number of contacts with increasing tilt angle can also be observed for the Cα AB loop and the Cβ A strand, CC' loop, and G strand (see *Figure 3—figure supplement 1*). Only for the Cα DE loop and Cβ EF loop, the average number of contacts is rather independent of the tilt angle. The tilt of the TCRαβ EC domain also affects the orientation of TM helices. The average inclination of the TCRα TM helix relative to the membrane normal decreases from about 11.5° to values around 8.5° with increasing tilt angle of the TCRαβ EC domain, while the average inclination of the CD3γ TM helix increases from about 15.5° to values around 24° (see *Figure 3(e)*). An increase from about 19° to values around 22° with increasing tilt angle of the TCRαβ EC domain also occurs for the average inclination angle of the TM helix of the ε chain of CD3εγ (see *Figure 3—figure supplement 2*). The average orientation of the other five TM helices relative to the membrane normal exhibits only small variations with the tilt angle of the TCRαβ EC domain.

## Discussion

The coupling of the tilt angle of the TCRαβ EC domain to overall conformational changes in the TCR – CD3 complex, which we observe in our MD simulations, provides insights on conformational changes induced by transversal, tilt-inducing forces. Transversal forces acting on the TCRαβ EC domain after binding to MHC-peptide-antigen complexes arise during the scanning of antigen-presenting cells by T cells (*Göhring et al., 2020*; *Cai et al., 2017*; *Huse, 2017*; *Rushdi et al., 2020*). While experiments indicate that the TCR – CD3 complex responds to mechanical force (*Kim et al., 2009*; *Feng et al., 2017*) and that the Cβ FG loop plays a key role in this response (*Das et al., 2015*), an outstanding question is how this force alters the conformation of the TCR – CD3 complex (*Courtney et al., 2018*). Our MD simulations show that an increased tilt of the TCRαβ EC domain leads to a marked decrease in the contacts between the Cβ FG loop and the CD3εγ EC domain (see *Figure 3(d)*), and also to changes in the inclination of TM helices relative to the membrane normal (see *Figure 3(e)*). Such structural changes in the TM domain of the TCR – CD3 have been suggested to be involved in the transmission of forces from the EC domain to the signaling motifs on the intracellular segments of the CD3 chains (*Brazin et al., 2015*; *Brazin et al., 2018*), which are not resolved in the cryo-EM structure. Similar to the Cβ FG loop, we also find a decrease of contacts between the Cα AB loop, which has been implicated in TCR triggering (*Beddoe et al., 2009*), and the ε chain of the CD3εγ EC domain with increasing tilt angle of the TCRαβ EC domain (see *Figure 3—figure supplement 1*).

Based on our simulation results for the orientational variations of the unbound TCR EC domain, the tilt of the bound TCR-MHC complex induced by a transversal force $f$ parallel to the membrane can be estimated under the assumption that the membrane anchoring of the MHC EC domain is

more flexible than the membrane anchoring of the TCR EC domain within the TCR - CD3 complex. This assumption appears reasonable because MHC class I and MHC class II EC domains are anchored by one and two peptide linkers, respectively, to a pair of transmembrane helices, whereas the three EC domains of the TCR - CD3 complex are jointly anchored by six linkers to a bundle of eight transmembrane helices. The anchoring flexibility of MHC class I molecules is also illustrated by a presumably binding-incompetent, supine conformation observed in two-dimensional crystals in which the MHC EC domains are positioned with their 'sides' on the membrane, rather than 'standing up' (*Mitra et al., 2004*). In the absence of transversal forces, the tilt-angle distribution of the TCR-MHC EC domain then can be approximated by the distribution of the TCR EC domain tilt angle observed in our simulations of the TCR – CD3 complex. From the energy contribution $-f\,h\sin[\tau]$ associated with a transversal force $f$ on the TCR-MHC EC complex with extension $h \simeq 13$ nm (*Wang et al., 2009*) along the tilt axis and tilt angle $\tau$ in units of rad, the maximum of the tilt-angle distribution can be estimated to be shifted from 34° for zero force to 41° for a transversal force $f = 2$ pN and to 49° for a transversal force $f = 5$ pN acting on the EC domain of the TCR-MHC complex (see *Figure 3—figure supplement 3(a)* and Materials and methods). Such transversal forces on TCR-MHC complexes up to 5 pN are within the range measured with force sensors (*Göhring et al., 2020*).

An increased tilt of the TCR-MHC complex due to transversal forces also affects the membrane separation at the site of the complex and, thus, the size-based segregation of large surface molecules such as CD45 or of other receptor-ligand complexes from TCR-MHC complexes. In the kinetic segregation mechanism, a key step in T cell activation is the size-based segregation of the inhibitory tyrosine phosphatase CD45 from TCR-MHC complexes (*Davis and van der Merwe, 2006*; *Choudhuri and van der Merwe, 2007*; *Chang et al., 2016*). A change of the tilt angle of the EC domain of the TCR-MHC complex with length $h \simeq 13$ nm from 34° to 49° leads to a decrease of about 2.2 nm in the separation $h\cos[\tau]$ between the membrane surfaces and, thus, to an increased segregation of large surface molecules. Such force-induced decreases of the membrane separation at the site of TCR-MHC complexes may also be relevant for the the recently observed segregation of CD2-CD58 complexes from TCR-MHC complexes, which both have EC domain lengths of about 13 nm (*Demetriou et al., 2019*).

The orientations of the TCRαβ EC domain within the TCR – CD3 complex are affected by the low-affinity interactions with the CD3εγ and CD3εδ EC domains (*He et al., 2015*). Because of the inherent limitations of coarse-grained models to describe such low-affinity interactions (*Javanainen et al., 2017*), we chose state-of-the-art atomistic force fields for our simulations of the TCR – CD3 complex. Our 120 simulation trajectories with a length of 1 μs provide a cumulative sampling on timescales that exceed the length of the individual trajectories (*Pande et al., 2003*; *Noé et al., 2009*) and lead to equilibrated conformational and orientational distributions of the EC domains (see *Figure 4*). The rather large orientational variations of the TCRαβ EC domain observed in our simulations make it plausible that processes on longer timescales do not contribute significantly to the overall EC domain conformations. Our sampling of the TM domain, in contrast, may be limited to conformations close to the cryo-EM structure of the TM domain, which is embedded in the detergent digitonin in the experiments. Larger conformational rearrangements of the TM helices such as the bending of the TCRα TM helix at a helix hinge observed in NMR experiments (*Brazin et al., 2018*) may occur on longer timescales, and may also depend on the composition of the lipid membrane. The TCRα TM helix remains intact on the microsecond timescales of our simulations and does not break into two helix halves connected by a hinge. Overall, our simulation result for the TM domain illustrate that the tilt of the TCRαβ EC domain is associated with statistically significant changes in the orientation of TM helices. How these orientational changes are affected by the membrane composition, and whether they can be related to conformational changes in the largely disordered intracellular signaling domains requires further simulations, likely with atomistic resolution because of limitations in modeling secondary structure propensities and disordered protein segments with coarse-grained force fields (*Monticelli et al., 2008*; *Robustelli et al., 2018*). In recent modeling based on the cryo-EM structure of the TCR – CD3 complex, the intracellular signaling domains have been included in coarse-grained simulations of the entire complex with a cumulative simulation time of 25 μs (*Prakaash et al., 2021*), and the conformations of the TM domain in a complex, asymmetric membrane have been explored in atomistic simulations with a cumulative simulation time of about 4 μs (*Lanz et al., 2021*). Future atomistic simulations of the TCR – CD3 complex

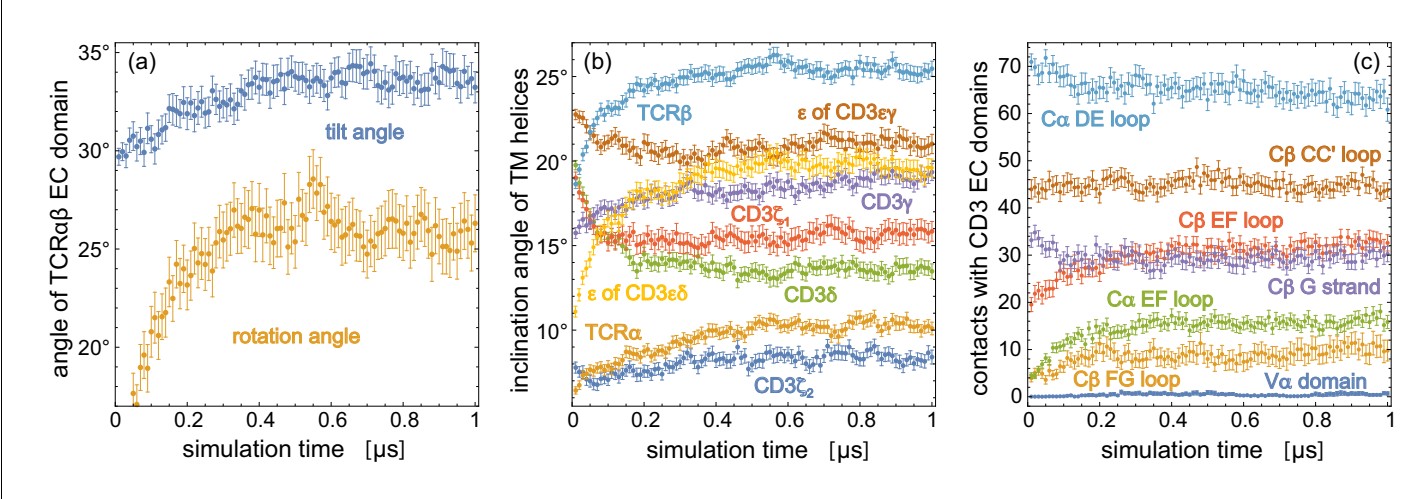

**Figure 4.** Time-dependent trajectory averages for (a) the tilt angle and rotation angle of the TCRαβ EC domain, (b) the inclination angles of the TM helices of the eight protein chains relative to the membrane normal, (c) the number of contacts of structural elements in the TCR constant domains Cα and Cβ and of the variable domain Vα with the two CD3 EC domains. Each data point is an average over the simulation structures of the 120 trajectories at the indicated time point, with error bars representing the error of the mean for these 120 structures. The structural elements of Cα and Cβ are defined in *Figure 2*.

The online version of this article includes the following figure supplement(s) for figure 4:

**Figure supplement 1.** Time-dependent trajectory averages for (a) the angle between the axes A and B of the TCRαβ EC domain and (b) minimal $C_\alpha$-atom root-mean-square deviations (RSMDs) of the TCRαβ, CD3εγ, and CD3εδ EC domains as well as the transmembrane (TM) domain relative to the cryo-EM structure of the T cell receptor – CD3 complex (*Dong et al., 2019*).

bound to the MHC-peptide EC domain may provide insights on the role of binding-induced conformational changes in TCR activation (*Hwang et al., 2020*; *Ayres et al., 2016*). The orientational distributions of the TCRαβ EC domain complex obtained from our simulations also provide a basis for the coarse-grained or multiscale modeling of the TCR-MHC complex anchored to apposing membranes (*Steinkühler et al., 2019*).

The concerted conformational changes of the TCR – CD3 complex in our simulations are reflected in the correlations of contact clusters in the EC domain interactions. The largest positive correlations of clusters in the TCRαβ/CD3εδ contact map of *Figure 2* occur between the contact clusters 1 to 4 (see *Figure 2—figure supplement 1*, top left), which correspond to interactions of the variable domain Vα of TCRα and the δ chain of CD3εδ. The large positive correlations of these four contact clusters can be understood from the coupling to the tilt of the TCRαβ EC domain, because the contacts of the Vα domain reported by the four contact clusters are only possible at high tilt angles of the TCRαβ EC domain (see *Figure 3(d)*). Relatively large positive correlations occur also between the clusters 13 to 21 of the TCRαβ/CD3εγ contact map (see *Figure 2—figure supplement 1*, bottom left). These clusters reflect interactions of the Cβ FG loop and G strand with the ϵ chain and the N-terminus of the γ chain of CD3εγ, which decrease with increasing tilt angle of the TCRαβ EC domain (see *Figure 3(d)* and *Figure 3—figure supplement 1*). Besides these positive correlations that result from the concerted conformational changes coupled to the tilt angle of the TCRαβ EC domain, the overall weak correlations between the majority of the other contact clusters in *Figure 2* indicate independent motions of the loops and strands involved in these EC domain contacts. In addition, relatively strongly negative correlations of several pairs of contact clusters point to alternative EC domain contacts. For example, the overall rather negative correlations between the clusters 13 to 21 and the cluster 22 of the TCRαβ/CD3εγ contact map show that the interaction of the Cβ G strand and the γ AB loop reported by cluster 22 is not compatible with the interactions of the Cβ FG loop and G strand to the ϵ chain and the N-terminus of the γ chain of CD3εγ, which are reflected by the clusters 13 to 21.

Overall, our simulations reveal that the orientation of the TCR EC domain relative to the membrane is coupled to structural changes throughout the TCR – CD3 complex. Besides this concerted

structural motion, the overall weak correlations of the majority of EC domain interactions indicate additional independent motions of the loops and strands that are involved in these interactions.

## Materials and methods

### System setup

To embed the cryoEM structure of the human TCR-CD3 complex (PDB ID 6jxr) into a lipid membrane, we have first aligned the protein complex along the $z$-axis of the simulation box. In this alignment with the program Visual Molecular Dynamics (VMD) (*Humphrey et al., 1996*), the first principal axis of the protein complex is parallel to the z-axis. We then embedded the aligned TCR-CD3 complex with the CHARMM-GUI program (*Wu et al., 2014*; *Jo et al., 2008*; *Lee et al., 2020*) into a lipid membrane that is oriented along the x-y-plane of the simulation box and is composed of 228 palmitoyl-2-oleoyl-sn-glycero-3-phosphocholine (POPC) and 57 cholesterol molecules in each monolayer, added missing atoms of the proteins with this program, and capped the N- and C-terminal ends of the eight protein chains with neutral ACE $(-COCH_3)$ and NME $(-NHCH_3)$ residues. We solvated this membrane-protein assembly at a salt concentration of 0.15 M KCl such that a 2.5 nm thick water layer is maintained above and below in $z$-direction. We performed the membrane embedding and solvation 10 times to obtain 10 system conformations as starting conformations of our simulations. The number of water molecules in this ten conformations slightly varies from 79,744 to 79,855.

### System equilibration

We have equilibrated the ten system conformations with the Amber16 software (*Case et al., 2017*). In this equilibration, we have first performed an energy minimization with 5000 minimization steps of steepest decent and subsequent 5000 steps of the conjugant gradient algorithm. The positions of backbone atoms of the EC and TM domains of the proteins were harmonically restrained in this minimization with a force constant of 10 kcal mol$^{-1}$ Å$^{-2}$. We have subsequently heated the systems in two simulation steps of 5 ps and 100 ps with harmonic restraints on all protein and lipid atoms: (1) from 0 K to 100 K at constant volume, and (2) from 100 K to 303 K at a constant pressure of 1 bar using the Berendsen barostat with anisotopic pressure coupling and a pressure relaxation time of 2 ps. In both heating steps, we used a Langevin thermostat with a collision frequency of 1 ps$^{-1}$, a MD integration time step of 2 fs, and a force constant of 10 kcal mol$^{-1}$ Å$^{-2}$ for the harmonic restraints. We have finally performed equilibration simulations with a total length of 20 ns at the temperature 303 K and a constant pressure of 1 bar. The equilibration simulations were carried out in ten steps of 2 ns with decreasing harmonic restraints on the protein backbone atoms of 10.0, 8.0, 6.0, 4.0, 2.0, 0.8, 0.6, 0.4, 0.2, and 0.1 kcal mol$^{-1}$ Å$^{-2}$ in these steps. We used a Langevin thermostat with collision frequency 1.0 ps$^{-1}$, a Berendsen barostat with a pressure relaxation time of 1 ps for the semi-isotropic pressure coupling, and in integration time step of 2 fs in these simulations. The lengths of all bond involving hydrogens were constrained with the SHAKE algorithm (*Miyamoto and Kollman, 1992*; *Ryckaert et al., 1977*), and a cutoff length of 1.0 nm was used in calculating the non-bonded interactions with the Particle Mesh Ewald (PME) algorithm *Essmann et al., 1995*; *Darden et al., 1993*.

### Production simulations

For each of the 10 equilibrated system conformations, we have generated 12 independent MD trajectories with a length of 1 µs at a temperature of 303 K. The total simulation time of these 120 production trajectories thus is 120 µs. We conducted the production simulations with the Amber99SB-ILDN protein force field (*Lindorff-Larsen et al., 2010*), the Amber Lipid14 membrane force field (*Dickson et al., 2014*), and the software AMBER 16 GPU (*Salomon-Ferrer et al., 2013*; *Le Grand et al., 2013*). We used a Langevin thermostat with a Langevin collision frequency of 1.0 ps$^{-1}$, and the Berendsen barostat with semi-isotropic pressure coupling and a relaxation time of 1 ps to apply a constant pressure of 1 bar in all directions at which the membrane is tensionless. As in the equilibration simulations, the lengths of all bond involving hydrogens were constrained with the SHAKE algorithm (*Miyamoto and Kollman, 1992*; *Ryckaert et al., 1977*), and a cutoff length of 1.0 nm was used in calculating the non-bonded interactions with the Particle Mesh Ewald (PME) algorithm (*Essmann et al., 1995*; *Darden et al., 1993*). In addition, we applied hydrogen mass repartitioning

(*Hopkins et al., 2015*) to all the hydrogen atoms on protein and lipids in the production simulations, which allowed to increase to the MD integration time step to 4 fs.

## Analysis of trajectories

Our analysis of the structural and orientational variability of the TCR - CD3 complex along the simulation trajectories is based on the residue-residue contacts of the TCRαβ, CD3εδ, and CD3εγ EC domains, on the tilt and rotation angle of the TCRαβ EC domain relative the membrane plane, and on the inclination angles of the TM helices relative the membrane. We take two residues from different EC domains to be in contact if the minimum distance between non-hydrogen atoms of the residues is smaller than 0.45 nm. Our tilt and rotation angles of the TCRαβ EC domain are determined from two characteristic axes in the domain: Axis A connects the centres of mass of Cαβ and Vαβ, where Cαβ is the dimer of the constant domains Cα and Cβ, and Vαβ is the dimer of the variable domains Vα and Vβ. Axis B connects the centres of mass of the variable domains Vα and Vβ. We define the tilt angle of the TCRαβ EC domain as the angle between axis A and the membrane normal, and the rotation angle as the angle between axis B and the normal of the plane spanned by axis A and the membrane normal. The rotation angle describes the rotation of the TCRαβ EC domain around axis A. To determine the inclination angles of the TM helices, we divide the residue span of the helices defined in the PDB file 6jxr of the cryo-EM structure (*Dong et al., 2019*) into two halves. We calculate the inclination angle of a TM helix as the angle between the membrane normal and the line that connects the centres of mass of the two helix halves.

*Figure 4* indicates that the structural and orientational ensembles sampled by our 120 production trajectories equilibrate within the first 0.5 μs of the trajectories. The data points in the figure represent averages over the 120 simulation structures of the trajectories at the indicated simulation times, with error bars representing the error of the mean for these 120 frames. The time-dependent trajectory averages for the tilt and rotation angle *Figure 4(a)* converge to average values of about 26° for the rotation angle and between 33° and 34° for the tilt angle within 0.5 μs. Within this time, the inclination angles of TM helices in *Figure 4(b)* and the numbers of contacts for structural elements in the TCR constant domains Cα and Cβ and for the variable domain Vα converge as well. We focus in our analysis therefore on the second trajectory halves and extract 50 structures at the simulations times 0.51 μs, 0.52 μs, 0.53 μs, ... , 1.0 μs from each trajectory, which results in $120 \times 50 = 6000$ structures that reflect an equilibrated ensemble of simulation conformations. The contacts, contact numbers, and angles presented in *Figures 2* and *3* are calculated from these 6000 structures. The 6000 structures are available at the Edmond Open Research Data Repository at https://dx.doi.org/10.17617/3.5m (*Pandey and Weikl, 2021*).

The variations of the tilt and rotation angle of the TCRαβ EC domain are large compared to the variations of the angle between the axes A and B of the EC domain. The tilt angle distribution of the equilibrated ensemble of 6000 simulation structures has a mean of 33.4° and a standard deviation of 8.9°, the rotation angle distributions has a mean of 26.1° and a standard deviation of 15.1°, while the distribution of the angle between axis A and B of the TCRαβ EC domain has a mean of 86.5° and a significantly smaller standard deviation of 1.9°. This small standard deviation indicates that the large TCRαβ EC domain can be seen as rather stable and rigid within the TCR - CD3 complex, which is supported by the small average root-mean-square deviation (RMSD) of about 2 Å for the $C_\alpha$ atoms of the TCRαβ EC domain simulation structures relative to the cryo-EM structure (see *Figure 4—figure supplement 1*). The average $C_\alpha$ RMSDs of the CD3εδ and CD3εγ EC domains relatively to the cryo-EM structure are only slightly larger and exhibit a marginal increase of about 0.3 Å along the second trajectory halves. Overall, these RMSDs and the small variations of the angle between axis A and B of the TCRαβ EC domain indicate that the three EC domains of the TCR - CD3 complex are rather stable. The main structural variations of the complex arise from the orientational variations of TCRαβ EC domain relative to the membrane, which are associated with the characteristic changes in the quaternary interactions between the EC domains illustrated in *Figure 3*.

We estimate the tilt-angle distributions of bound TCR-MHC complexes that experience a transversal force $f$ under the assumption that the membrane anchoring of the MHC EC domain is more flexible than the membrane anchoring of the TCR EC domain. The force-free tilt angle distribution of the TCR-MHC EC domain then can be approximated by the distribution $P_0(\tau)$ for the tilt angle $\tau$ of the TCR EC domain obtained from our simulations of the TCR - CD3 complex. From this distribution, an effective free energy $E_0(\tau)$ associated with the tilt in the absence of transversal force can be

estimated *via* $P_0(\tau) = \exp[-E_0(\tau)/k_B T]$ with Boltzmann constant $k_B$. A transversal force $f$ parallel to the membrane shifts the effective energy to $E_f(\tau) = E_0(\tau) - f\,h\sin[\tau]$ where $h \simeq 13$ nm is the length of the TCR-MHC EC domain (*Wang et al., 2009*) along the tilt axis, and $\tau$ denotes the tilt angle in units of rad. This transversal force is transmitted by the T cell cytoskeleton *via* a frictional coupling to the intracellular side of the TCR – CD3 complex according to experiments of T cell adhesion to patterned supported membranes (*DeMond et al., 2008*; *Mossman et al., 2005*). We assume that the TCR-MHC complex rotates and aligns its tilt direction to the direction of the force. The tilt-angle distribution under a transversal force $f$ then follows as $P_f(\tau) = \exp[-E_f(\tau)/k_B T]/\int \exp[-E_f(\tau)/k_B T]\mathrm{d}\tau$ (see *Figure 3—figure supplement 3(a)*). This distribution thus is obtained by multiplication of $\exp[f\,h\sin[\tau]/k_B T]$ to the force-free distribution and subsequent normalization.

We determined the membrane thickness in *Figure 4—figure supplement 1(b)* as the thickness of the annulus of POPC lipids in contact with the TM domain. Here, POPC lipids are taken to be in contact with a protein chain if the minimum distance between the non-hydrogen atoms of the lipid and protein chain is smaller than 0.5 nm. The thickness of the POPC lipid annulus in a simulation conformation then was calculated as the distance between the centres of mass of the POPC lipid headgroups in the two monolayers of the annulus along the direction perpendicular to the membrane. Within the statistical accuracy, the membrane thickness does not change with increasing tilt angle of the TCRαβ EC domain.

## Additional information

### Funding
No external funding was received for this work.

### Author contributions
Prithvi R Pandey, Software, Formal analysis, Validation, Investigation, Methodology, Writing - original draft; Bartosz Różycki, Formal analysis, Methodology; Reinhard Lipowsky, Conceptualization, Supervision, Funding acquisition; Thomas R Weikl, Conceptualization, Formal analysis, Supervision, Investigation, Visualization, Methodology, Writing - original draft

### Author ORCIDs
Bartosz Różycki (iD) https://orcid.org/0000-0001-5938-7308
Reinhard Lipowsky (iD) http://orcid.org/0000-0001-8417-8567
Thomas R Weikl (iD) https://orcid.org/0000-0002-0911-5328

### Decision letter and Author response
Decision letter https://doi.org/10.7554/eLife.67195.sa1
Author response https://doi.org/10.7554/eLife.67195.sa2

## Additional files
### Supplementary files
• Transparent reporting form

### Data availability
All 6000 molecular dynamics structures of the membrane-embedded TCR-CD3 complex used in the analysis have been deposited in the Edmond Open Research Data Repository under https://doi.org/10.17617/3.5m.

The following dataset was generated:

| Author(s) | Year | Dataset title | Dataset URL | Database and Identifier |
|---|---|---|---|---|
| Pandey PR, Weikl TR | 2021 | MD simulation structures of the membrane-embedded TCR - CD3 complex | https://doi.org/10.17617/3.5m | Edmond Open Research Data Repository of the Max |

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
