## [Decision Letter]

**Acceptance summary:**

Your work shows, based on Molecular Dynamics simulations, the structural variability of the T cell receptor complex. Changes in the tilt angles of the extracellular antigen binding domains were observed to correlate with changes in the orientation of transmembrane helices, which may well impact signal transduction during antigen recognition. Your responses to the reviewers have improved the clarity of the approach and interpretations are well supported by experimental observations where available.

**Decision letter after peer review:**

Thank you for submitting your article "Structural variability and concerted motions of the T cell receptor – CD3 complex" for consideration by *eLife*. Your article has been reviewed by 3 peer reviewers, including Michael L Dustin as the Reviewing Editor and Reviewer #1, and the evaluation has been overseen by Tadatsugu Taniguchi as the Senior Editor. The following individual involved in review of your submission has agreed to reveal their identity: Brian Baker (Reviewer #2).

Essential revisions:

All the reviewers found the early results on MD simulations of full TCR to be of great interest and potential value. The paper currently doesn't provide sufficient insight for publication in *eLife*, but it was felt that additional discussion could address this in different directions that might be best selected by the authors. The reviewers didn't reach a consensus, but hope that with the individual reviews below the authors can revise the manuscript to increase the value for immunology and signalling audiences along one or two of the lines suggested.

1) There was some consensus regarding discussion of the limitations of the short simulations with implications for stability of the system and the ability to establish correlations. Is there potential to use the results from these simulations to develop coarser simulations over longer time that might get into the time frame of pMHC interaction and force application. A discussion of these future directions could be helpful in addition to pointing out current limitations.

2) A video showing the TCR fluctuations would be nice.

*Reviewer #1:*

It would be important to determine if this atomistic simulation lasting 1 µs could be used to seed a coarse grained simulation that could operate in time frames relevant to natural ligand binding and capture the major movements documented here, for example the 4 clusters, to enable MD simulations of sufficient duration to ask questions about TCR signalling in a realistic time frame.

*Reviewer #2:*

While the strengths of the paper are in the testable hypotheses that are generated, there are weaknesses that should be considered:

– The simulations give a window into thermal fluctuations around the 'average' cryoEM structure. The extent these rapid motions give insight into signaling mechanisms is limited. For example, there is no comparison of how the motions of a pMHC bound structure might differ, or how fluctuations might be altered under load.

– The authors do a limited analysis of equilibration, which is always needed in complex simulation papers to ensure the robustness of the data and conclusions.

– There is a limited analysis of structural variance or correlated motion. Overall, the authors give very limited attention to the high level of detail that MD simulations are capable of and arguably best known for.

Comments for the authors:

There are major weaknesses that should be considered:

1) The authors perform a total of 120 microseconds of simulation in explicit solvent, performed by compositing many shorter simulations. This is a considerable amount of simulation time. However, the authors are still looking at motions that would be comparable to experiments on the nanosecond timescale. It is highly unlikely that these simulations would capture what occurs upon binding or applied force, which would occur with higher barriers and over longer timescales. Instead, we are looking at thermal fluctuations around an equilibrium structure. While still providing testable hypotheses regarding how a TCR/CD3 might be 'poised' for signaling, the immediate insight into signaling mechanisms and thus impact is very limited. The authors need to consider this throughout their manuscript.

2) The authors do not do a complete analysis of equilibration, using domain angles and contacts as a window into equilibration. There are none of the analyses that are traditionally performed with long simulations to ensure equilibration of the structure (e.g., is domain assembly maintained, how is secondary or tertiary structure maintained, what about membrane stability, etc.).

3) Similar to the point above, the analysis is limited to contacts and angles. One might expect various higher frequency motions to be insightful – for example, what does the structure of the FG loop do over the course of the simulation? That about the β chain AB loop, which has been implicated in triggering? The overall analysis is very high level and lacking in the kind of rich detail that extensive MD simulations are capable of.

4) There are no direct connections to experiments here. Experimental data do not need to be included, but over the years there have been many mutation, perturbations, etc. performed that the authors could look at. Similarly, there are no pMHC bound or force experiments included that could give insight into actual signaling mechanisms as opposed to the ligand-free and force-free fluctuations that presumably occur as the molecule is waiting for something to happen.

5) Related to the point above, there is data suggesting dynamic allostery as a mechanism contributing to TCR triggering. Dynamic allostery requires correlated motion – none of that is considered here.

*Reviewer #3:*

Weikl and colleagues used the structure of the T cell receptor complex, which has recently been determined via cryo-electron microscopy, as basis for an atomistic modelling approach. This method offers the advantage to overcome a central limitation of cryo-EM, which is the choice of the membrane lipid environment: while experiments have been based on embedding the protein in detergent, Weikl et al. used here glycerophospholipids and cholesterol, which reflects the natural situation in the plasma membrane more appropriately. In addition, cryo-EM required the addition of fixatives, which is not necessary in the MD simulation approach.

The paper reveals interesting new dynamical information about the TCR complex. It would be informative, if the authors would include a discussion on the following points:

Figure 2: How is contact between residues defined? Would an isolated 10ns encounter already qualify as contact? What about analyzing the contact duration? What is distance between two sites to qualify as contact?

Figure 3a/b:

• It would be helpful to indicate rotation angle 0; maybe by adding an en face view onto the axis A?

• The tilting of the TM helices appears to be accompanied by slight local thinning of the membrane. Is that correct? Do lipids adjacent to the transmembrane helices follow the tilt, and/or is there different ordering of the fatty acids? Is the cholesterol distribution affected by the tilt? How would different lipids with different length or compressibility affect the helix tilting?

• What would generally happen if different lipids were tested, particularly asymmetric lipid distributions across the membrane? In the natural plasma membrane environment lipids are distributed asymmetrically across the leaflets, with saturated and unsaturated lipids of different chain length being enriched in the extracellular and cytoplasmic leaflet, respectively. It would be interesting, whether this compensates or probably even amplifies the observed mechanism. Maybe the authors could add a discussion on this aspect.

Figure 3 c and e: It would be informative to add the results of the cryo-EM study here.

Figure 3: For better comparison, it would be nice to scale the y-axes with identical increments.

If fluctuations of the TCR α/β would be similar in reality as it was revealed in the simulation, I would expect continuous fluctuations of helix tilt angles. If helix tilt angle was indeed a cause for signaling, wouldn't that lead to continuous aberrant activation of the TCR?

[Editors' note: further revisions were suggested prior to acceptance, as described below.]

Thank you for submitting your article "Structural variability and concerted motions of the T cell receptor – CD3 complex" for consideration by *eLife*. Your article has been reviewed by 2 peer reviewers, including Michael L Dustin as the Reviewing Editor and Reviewer #1, and the evaluation has been overseen by Tadatsugu Taniguchi as the Senior Editor.

The paper is significantly improved by the inclusion of the videos and discussion of some biological implications.

1) Is it correct that the rotation angle 0 is defined by the origination in the published cry-EM structure? Regardless, this should be defined more clearly in the text and figure.

2) The authors should address the points raised by reviewer 3 regarding force induced tilt through clarification of the text and explanatory schematics if helpful. It may also be interesting in addressing the last comment to determine if the observation of supine orientation of MHC class I at a membrane surface is relevant to the discussion. see Mitra AK, Celia H, Ren G, Luz JG, Wilson IA, Teyton L. Supine orientation of a murine MHC class I molecule on the membrane bilayer. Curr Biol. 2004;14(8):718-24. Epub 2004/04/16. doi: 10.1016/j.cub.2004.04.004. PubMed PMID: 15084288. Is this natural orientation of MHC class I aligned with the tilt of the TCR when the interface is formed? Does the tilt angle of the TCR create a natural rudder to orient the TCR and would it matter which of the CD3 or zeta-zeta tails are pulled.

*Reviewer #3:*

In principle, all of my previous questions were adequately addressed. There was a misunderstanding concerning my previous comment on the specification of the rotational angle in Figure 3: My problem was to understand, which TCR conformation corresponds to a rotation angle 0. The authors may still consider to add this information.

Concerning the new data on force-induced tilt, however, I have a few questions:

– First, the authors mention on multiple locations in their paper a force-induced tilt of the TCR-MHC complex. The MHC, however, was not included in their simulations. I suggest being more precise in this aspect.

– Second, if I understand correctly, force was not included in the simulations, but instead the effect was added a posteriori. I had difficulties to understand the rationale behind it. What is the justification for the equation given in line 346? What was actually multiplied by the exponential function?

– Third, wouldn't one expect a directionality of the effect? In other words, if force acted, say, in the opposite direction to the naturally occurring tilt, is the idea that the TCR would align with the external force field?

– Fourth, I would be more careful with speculations concerning CD45 segregation. The authors argue in the discussion (line 175 and following) that TCR tilt brings the two membranes in closer juxtaposition. But that would only be true if MHC would also be sufficiently flexible to compensate for the TCR tilt, keeping the two membranes parallel.

---

## [Author Response]

Essential revisions:All the reviewers found the early results on MD simulations of full TCR to be of great interest and potential value. The paper currently doesn't provide sufficient insight for publication in eLife, but it was felt that additional discussion could address this in different directions that might be best selected by the authors. The reviewers didn't reach a consensus, but hope that with the individual reviews below the authors can revise the manuscript to increase the value for immunology and signalling audiences along one or two of the lines suggested.

We first would like to thank all reviewers for their constructive and helpful comments and suggestions. To address these comments, we have added 3 new page-wide figures with subfigures and 3 videos and have substantially extended the text, in particular in the Discussions and Methods sections. To increase the value for immunology and signalling audiences:

– We have modelled the tilt of the TCR EC domain under a transversal force acting on the TCR – MHC complex, based on the force-free tilt-angle distribution TCR EC domain obtained from our simulations (see Figure 3 – supplement 3 (a)), and

– We discuss implications for size-based segregation that result from a decreased membrane separation due to a force-induced tilt of the TCR – MHC complex (see new paragraph starting on line 175 of the Discussions section).

1) There was some consensus regarding discussion of the limitations of the short simulations with implications for stability of the system and the ability to establish correlations. Is there potential to use the results from these simulations to develop coarser simulations over longer time that might get into the time frame of pMHC interaction and force application. A discussion of these future directions could be helpful in addition to pointing out current limitations.

We discuss simulation lengths, coarse-grained *versus* atomistic simulations, and future directions now in a new paragraph starting on line 187 in the Discussions section. In the last years, advances in computing on graphics processing units (GPUs) have made it possible to investigate protein association with atomistic models. We see it as a main contribution of our manuscript to make use of these advances in our atomistic simulations of the TCR – CD3 complex. As emphasized in the new paragraph of the Discussions section, we think that our simulations provide equilibrated conformational ensembles of the EC domain arrangements of the TCR – CD3 complex. We also use popular coarse-grained models such as the MARTINI model in simulations of other, larger systems in our group, but think that these coarse-grained models are not reliable for weak-affinity complexes or disordered segments because of inherent limitations regarding binding affinities and secondary structure propensities. In the MARTINI model, for example, secondary structure needs to be predefined and constrained, and also tertiary structure stability typically requires further artificial constraints. But we also mention that the orientational distributions of the TCR – CD3 complex obtained from our atomistic simulations can be used for multiscale modeling of TCR – MHC complexes anchored to apposing membranes, as in our recent multiscale modeling of CD47 – SIRPa complexes in Steinkühler et al., 2019. We also show now that the EC domains are stable in our simulations (see Figure 4 – supplement 1 and new text paragraph starting on line 327 of Methods section), which confirms that the main structural variations of the TCR – CD3 complex discussed in the Results section arise from the orientational variations of TCRab EC domain and the changes in the quaternary interactions between the EC domains associated with these orientational variations.

2) A video showing the TCR fluctuations would be nice.

We agree, and have added three videos of trajectory segments along which the tilt angle increases from around 20° to values above 50°, together with the new Figure 3 – supplement 4 showing the values of the tilt and rotation angle along these trajectory segments.

Reviewer #1:It would be important to determine if this atomistic simulation lasting 1 µs could be used to seed a coarse grained simulation that could operate in time frames relevant to natural ligand binding and capture the major movements documented here, for example the 4 clusters, to enable MD simulations of sufficient duration to ask questions about TCR signalling in a realistic time frame.

Please see our response to the essential point (1) above. We think that we have “solved the sampling problem” in our atomistic simulations, at least for the EC domain arrangements of the TCR – CD3 complex. In other words, we think that our results for the orientation and quaternary interactions of the TCR EC domain within the TCR – CD3 complex are not limited by the (cumulative) simulation times. And we think that the accuracy of atomistic models is necessary here, because only state-of-the-art atomistic models are sufficiently reliable in reproducing stable, folded tertiary structures of proteins and quaternary interactions of proteins. Overall, we think that a main remaining challenge for TCR signalling is to include the largely disordered intracellular signaling domains in atomistic simulations. These signaling domains are not included in the cryo-EM structure, and therefore beyond the scope of this manuscript.

Reviewer #2:[…] There are major weaknesses that should be considered:1) The authors perform a total of 120 microseconds of simulation in explicit solvent, performed by compositing many shorter simulations. This is a considerable amount of simulation time. However, the authors are still looking at motions that would be comparable to experiments on the nanosecond timescale. It is highly unlikely that these simulations would capture what occurs upon binding or applied force, which would occur with higher barriers and over longer timescales. Instead, we are looking at thermal fluctuations around an equilibrium structure. While still providing testable hypotheses regarding how a TCR/CD3 might be 'poised' for signaling, the immediate insight into signaling mechanisms and thus impact is very limited. The authors need to consider this throughout their manuscript.

We argue now in the new paragraph starting on line 187 of the Discussions section that our 120 trajectories provide a cumulative sampling on timescales that exceed the length of the individual trajectories (see new references Pandey et al., 2003 and Noe et al. 2009). We further argue that this cumulative sampling on microsecond timescale leads to equilibrated conformational and orientational distributions of the EC domains, as shown in Figure 4. We comment that the rather large orientational variations of the TCRab EC domain EC domain observed in our simulations make it plausible that processes on longer timescales do not contribute significantly to the overall EC domain conformations. Based on the equilibrated tilt-angle distribution determined from our simulations, we now estimated how transversal forces of 2 or 5 pN, which are in the range of experimentally determined transversal forces on TCR – MHC complexes, shift the tilt-angle distributions (see new Figure 3 – supplement 3, and Discussions and Methods section). We now discuss also the implications of force-induced tilt for length-based segregation, which is a key element of the kinetic segregation mechanism of T cell activation (see new paragraph starting on line 175 of the Discussion section).

2) The authors do not do a complete analysis of equilibration, using domain angles and contacts as a window into equilibration. There are none of the analyses that are traditionally performed with long simulations to ensure equilibration of the structure (e.g., is domain assembly maintained, how is secondary or tertiary structure maintained, what about membrane stability, etc.).

We have added a stability analysis of the EC domains and the TM domain in Figure 4 supplement 1. The analysis shows stability on the level of tertiary structures, i.e. stability of EC domains, and that the dominant variations analysed in the main Figures 2 and 3 are variations in the quaternary structure and orientation of the TCR – CD3 complex.

3) Similar to the point above, the analysis is limited to contacts and angles. One might expect various higher frequency motions to be insightful – for example, what does the structure of the FG loop do over the course of the simulation? That about the β chain AB loop, which has been implicated in triggering? The overall analysis is very high level and lacking in the kind of rich detail that extensive MD simulations are capable of.

The contact probabilities in our main Figure 2 provide a highly detailed, but still “human readable” representation of the quaternary interactions between the CD3 domains in our equilibrated simulations. We emphasize in the manuscript that the Cb FG loop loses quaternary contacts with increasing tilt angle of the TCR EC domain. We thank the referee for pointing out the role of the Ca AB loop in triggering (assuming that this is the loop the referee refers to). We find that the Ca AB loop also clearly loses quaternary contacts with increasing tilt angle of the TCR ECab domain (see Figure 3 – supplement 1), similar to the Cb FG loop, and address this now on lines 164 to 167 in the first pagraph of the Discussions section. Based on our simulations, changes in the Ca AB loop structure thus can be expected if TCR binding and activation is associated with an increased tilt of the TCRab EC domain, because changes in quaternary contacts of a loop also affect loop structure. Because of complexity, a detailed comparison to the fluorescence-based and mutational experiments regarding the Ca AB loop in Beddoe et al., 2009, is beyond the scope of our manuscript. We agree with the referee that there may well be further details in our simulations regarding e.g. loop structure that deserve attention in other contexts. Also for this reason, we have published all 6000 simulation structures on which our analysis is based (see Pandey and Weikl, 2021) for further analysis by others.

4) There are no direct connections to experiments here. Experimental data do not need to be included, but over the years there have been many mutation, perturbations, etc. performed that the authors could look at. Similarly, there are no pMHC bound or force experiments included that could give insight into actual signaling mechanisms as opposed to the ligand-free and force-free fluctuations that presumably occur as the molecule is waiting for something to happen.

We agree with the referee that experimental data from, e.g. mutational experiments provided indirect insights on the structure of the TCR – CD3 complex. We refer to essential data of mutational and other experiments in the first paragraph of the Introduction from lines 35 on. With the publication of the cryo-EM structure of the TCR – CD3 complex as a breakthrough in 2019, this cryo-EM structure has of course become the main reference regarding the structure of the complex. We therefore focus in our analysis on a detailed comparison to the cryo-EM structure (see Figures 1 and 2). We see the main point of our manuscript in an atomistically detailed, extensive computational analysis that goes beyond unavoidable limitations of the cryo-EM structure due to the embedding in the detergent digitonin and the chemical crosslinking of the protein chains.

5) Related to the point above, there is data suggesting dynamic allostery as a mechanism contributing to TCR triggering. Dynamic allostery requires correlated motion – none of that is considered here.

We now refer to binding-induced conformational changes such as dynamic allostery as an outlook on lines 213 to 216 of the Discussion section. Correlations of quaternary interactions in our simulation of the unbound TCR – CD3 complex are analysed in detail in Figure 1 – supplement 1.

Reviewer #3:[…] The paper reveals interesting new dynamical information about the TCR complex. It would be informative, if the authors would include a discussion on the following points:Figure 2: How is contact between residues defined? Would an isolated 10ns encounter already qualify as contact? What about analyzing the contact duration? What is distance between two sites to qualify as contact?

We define contacts based on distances between non-hydrogen atoms in the cryoEM structure and simulation conformations, and state the contact definition now also in the main text of the Results section:

“As in the contact maps for the cryo-EM structure shown in Figure 1, two residues are taken to be in contact in a simulation structure if the minimum distance between non-hydrogen atoms of the residues is smaller than 0.45 nm.” (lines 83 to 86).

This standard, “instantaneous” definition of contacts is both applicable to the cryo-EM structure and our simulation confirmations. An isolated 10 ns encounter can lead to contacts according to this definition. But in Figure 2 and throughout our analysis, we only include contacts that occur on at least 0.5% of the 6000 simulation conformations on which our analysis is based – i.e. we only include contacts that occur on at least 30 simulation confirmations, irrespective of contact duration, and on how many trajectories the contact occur. We give as example:

“Residue-residue contacts between Va and the d chain of CD3ed have probabilities smaller than 3%, but occur in 75 of the 120 trajectories and are, thus, a robust feature of our simulations.” (lines 100 to 102).

In general, also stable contacts typically “flicker” on MD trajectories due to structural fluctuations, which complicates temporal analyses of contact durations.

Figure 3a/b:• It would be helpful to indicate rotation angle 0; maybe by adding an en face view onto the axis A?

The rotation angle 0 is a lower limit of rotation angles occurring in our simulation conformations. But we think that our 3 new videos, together with the new Figure 3 – supplement 4, in which the tilt and rotation angles along the videos are shown, address the point of the referee to better illustrate the rotational motion. For example, the videos 1 and 2 show that an increase of the rotation angle with the tilt angle brings the Va loops in contact with CD3ed. Such an increase of the rotation angle with increasing tilt is “typical” because of the correlation of the angles illustrated in Figure 3(c). Video 3 shows an “atypical” situation in which the rotation angle does not increase with the tilt angle.

• The tilting of the TM helices appears to be accompanied by slight local thinning of the membrane. Is that correct? Do lipids adjacent to the transmembrane helices follow the tilt, and/or is there different ordering of the fatty acids? Is the cholesterol distribution affected by the tilt? How would different lipids with different length or compressibility affect the helix tilting?

This is an interesting question that we address in the new Figure 3 – supplement 3 (b). The answer is: Within the statistical accuracy, there is no change in the thickness of the membrane annulus around the TM domain with increasing tilt angle. A definition of this membrane annulus as lipids in contact with protein chains is given on lines 350 to 357 of the Methods section. We also don’t see any changes in the composition (POPC versus cholesterol) in this membrane annulus with changing tilt (data not shown).

• What would generally happen if different lipids were tested, particularly asymmetric lipid distributions across the membrane? In the natural plasma membrane environment lipids are distributed asymmetrically across the leaflets, with saturated and unsaturated lipids of different chain length being enriched in the extracellular and cytoplasmic leaflet, respectively. It would be interesting, whether this compensates or probably even amplifies the observed mechanism. Maybe the authors could add a discussion on this aspect.

We agree that the composition of the membrane is of relevance for changes in the TM domain. We have added now on lines 202 and following in the Discussions section:

“Overall, our simulation result for the TM domain illustrate that the tilt of the TCRab EC domain is associated with statistically significant changes in the orientation of TM helices. How these orientational changes are affected by the membrane composition, and whether they can be related to conformational changes in the largely disordered intracellular signaling domains requires further simulations, likely with atomistic resolution because of limitations in modelling secondary structure propensities and disordered protein segments with coarse-grained force fields.”

Figure 3 c and e: It would be informative to add the results of the cryo-EM study here.

This is an interesting but also tricky point. Because the cryo-EM structure does not contain a lipid membrane, the orientation of this structure in a membrane is unclear. In addition, it is unclear how the chemical crosslinking in the structural experiments affects the orientation of the EC domains relative to the TM domain. We have used structural alignment (superposition) to the TM domain of our simulation structures to embed and orient the cryo-EM structure in a membrane. From alignment to a large number of simulation structures, we obtain the estimates 31.2 ± 0.4° for the tilt angle and 14.7± 0.7° for the rotation angle of the cryo-EM structure. This is described now on lines 122 and following of the Results section. Strictly speaking, these are not “pure” experimental results, because they are based on structural alignments to our simulation structures. We therefore prefer to state these results in the text with explanations. We think that inclination helices of tilt angles in the cryo-EM structure are somewhat dubious even in our structural alignments, because of the missing membrane in the structural experiments. We therefore focus on the tilt and rotation angle.

Figure 3: For better comparison, it would be nice to scale the y-axes with identical increments.

We chose different sections and increments along the y-axes because the changes in contact numbers and inclination angles occur in different ranges. In other words, we think that the chosen y-axes sections are required for a clear representation of the data.

If fluctuations of the TCR α/β would be similar in reality as it was revealed in the simulation, I would expect continuous fluctuations of helix tilt angles. If helix tilt angle was indeed a cause for signaling, wouldn't that lead to continuous aberrant activation of the TCR?

We understand the question, and think that answering this question requires further simulations, as stated now on lines 204 and following of the Discussions section, because the intracellular signaling domains are not included in the cryo-EM structure and in our simulations, which are based on this structure.

[Editors' note: further revisions were suggested prior to acceptance, as described below.]

The paper is significantly improved by the inclusion of the videos and discussion of some biological implications.1) Is it correct that the rotation angle 0 is defined by the origination in the published cry-EM structure? Regardless, this should be defined more clearly in the text and figure.

To clarify this point, we have added on lines 114 and 115:

“A rotation angle of 0° indicates a TCRαβ EC domain orientation in which the centres of mass of the variable domains Vα and Vβ are equally close to the membrane”.

This statement follows from the definition of the rotation angle given on lines 110 and 111. The definition is based on the membrane normal and the two axes A and B of the TCRαβ EC domain defined on lines 106 to 109 and illustrated in Figure 3(b). At a rotation angle of 0°, the axis B that connects the centres of mass of the variable domains Vα and Vβ is parallel to the membrane.

The tilt and rotation angle of the TCRαβ EC domain in the cryo-EM structure cannot be determined directly, because of the missing membrane embedding. Without membrane embedding, the membrane normal is unclear. On lines 126 to 132 added in the previous, first revision, we describe how we estimate the tilt and rotation angle of the TCRαβ EC domain in the cryo-EM structure based on an alignment of the TM domain of this structure to our simulation conformations. This alignment orients the cryo-EM structure relative to our simulated membranes, and allows to calculate the orientation angles. We obtain a rotation angle of 14.7 ± 0.7° for the cryo-EM structure from this calculation. We have added now also on lines 115 to 117 that such positive rotation angles indicate “conformations in which the variable domain Vα is closer to the membrane than the variable domain Vβ“, see also Figure 3(a) and (b) – the conformations shown in this Figure have rotation angles of 12.8° and 42.9°.

2) The authors should address the points raised by reviewer 3 regarding force induced tilt through clarification of the text and explanatory schematics if helpful.

We have substantially extended the text both in the Discussions and Methods section, see lines 172 to 189 and lines 360 to 375. We now state also in the Discussions section that our estimation of the force-induced tilt assumes “that the membrane anchoring of the MHC EC domain is more flexible than the membrane anchoring of the TCR EC domain”. The calculation is described in detail on lines 360 to 375 of the Methods section.

In essence, the calculation makes use of the standard statistical-physical Boltzmann factor to relate tilt-angle distributions to effective free energies for the tilt. We now also state on lines 371 to 372:

“We assume that the TCR-MHC complex rotates and aligns its tilt direction to the direction of the force.“

It may also be interesting in addressing the last comment to determine if the observation of supine orientation of MHC class I at a membrane surface is relevant to the discussion. see Mitra AK, Celia H, Ren G, Luz JG, Wilson IA, Teyton L. Supine orientation of a murine MHC class I molecule on the membrane bilayer. Curr Biol. 2004;14(8):718-24. Epub 2004/04/16. doi: 10.1016/j.cub.2004.04.004. PubMed PMID: 15084288. Is this natural orientation of MHC class I aligned with the tilt of the TCR when the interface is formed?

Strictly speaking, addressing this question requires to model the TCR-MHC complex, which is beyond the scope of our manuscript. However, Mitra et al. state that the supine orientation of the MHC class places “the peptide binding groove approximately perpendicular to the membrane surface.” We therefore find it plausible to say to that this conformation is “presumably binding-incompetent”, as we state now on line 180. The supine conformation may be stabilized by the two-dimensional crystal of the experiments. We think that the supine conformation can be seen to indicate high anchoring flexibility. We now state on lines 179 to 182:

“The anchoring flexibility of MHC class I molecules is also illustrated by a presumably binding-incompetent, supine conformation observed in two-dimensional crystals in which the MHC EC domains are positioned with their "sides" on the membrane, rather than "standing up" (Mitra et al., 2004).”

Does the tilt angle of the TCR create a natural rudder to orient the TCR and would it matter which of the CD3 or zeta-zeta tails are pulled.

This is an interesting question, which leads to the follow-up question: What controls the rudder? We tend to think that there is no “hand on the rudder”. So, without force, there are just (thermal) fluctuations of the orientation and tilt of the TCR EC domain. And with force on the bound TCR-CD3 complex, the rudder follows the drag – this is also implied by our assumption that the tilt direction aligns with the force direction, which is now stated on lines 371/72.

We also tend to think that it should not matter which intracellular CD3 segments are coupled to the T cell cytoskeleton. We think that a transversal, membrane-parallel force on any of these segments should lead to a dragging of the TM domain of 8 helices along the membrane – and a dragging and transport of the whole complex. This view is largely influenced by the experiments of DeMond et al., 2008 and Mossman et al., 2005, which indicate a “frictional coupling” between T cell cytoskeleton and TCR – CD3 complex, as included now on lines 369 to 371.

Reviewer #3:In principle, all of my previous questions were adequately addressed. There was a misunderstanding concerning my previous comment on the specification of the rotational angle in Figure 3: My problem was to understand, which TCR conformation corresponds to a rotation angle 0. The authors may still consider to add this information.

We indeed misunderstood this point. As indicated already above in our response to the suggestions of the reviewing editor, we have added on lines 114 and 115:

“A rotation angle of 0° indicates a TCRαβ EC domain orientation in which the centres of mass of the variable domains Vα and Vβ are equally close to the membrane”.

This statement follows from the definition of the rotation angle given on lines 110 and 111. The definition is based on the membrane normal and the two axes A and B of the TCRαβ EC domain defined on lines 106 to 109 and illustrated in Figure 3(b). At a rotation angle of 0°, the axis B that connects the centres of mass of the variable domains Vα and Vβ is parallel to the membrane.

Concerning the new data on force-induced tilt, however, I have a few questions:– First, the authors mention on multiple locations in their paper a force-induced tilt of the TCR-MHC complex. The MHC, however, was not included in their simulations. I suggest being more precise in this aspect.

We have extended and clarified the description of our modelling of force-induced tilt, both in the Discussions section (lines 172 to 189) and the Methods section (lines 360 to 375). In the first sentence on lines 172 to 175 that mentions this modelling, we state that the modelling is “based on our simulation results for the orientational variations of the unbound TCR EC domain”.

– Second, if I understand correctly, force was not included in the simulations, but instead the effect was added a posteriori. I had difficulties to understand the rationale behind it. What is the justification for the equation given in line 346? What was actually multiplied by the exponential function?

This is correct. We now provide a detailed description of the calculation on lines 360 to 375 of the Methods section. We agree that this description was too short in the previous version of the manuscript. The two main and now clearly stated assumptions of our calculations are (1) that the MHC complex is more flexibly anchored than the TCR EC domain, and (2) that the tilt direction of the TCR-MHC complex aligns with the direction of the transversal, membrane-parallel force. Our calculation then is largely based on the standard exponential Boltzmann factor to relate probability distributions to energies.

– Third, wouldn't one expect a directionality of the effect? In other words, if force acted, say, in the opposite direction to the naturally occurring tilt, is the idea that the TCR would align with the external force field?

Yes, this is correct. We now state on lines 371/72:

“We assume that the TCR-MHC complex rotates and aligns its tilt direction to the direction of the force.”

We think that this assumption is plausible also because of the experiments of DeMond et al., 2008 and Mossman et al. 2005 mentioned now on lines 369 to 371, which indicate a frictional coupling that “allows slip” (DeMond et al. 2008). In other words, we think that these experiments indicate that the coupling between T cell cytoskeleton and TCR – CD3 complex does not block or preclude a rotation of the TCR – CD3 complex. A rotation leading to alignment then is a consequence of the exerted force.

– Fourth, I would be more careful with speculations concerning CD45 segregation. The authors argue in the discussion (line 175 and following) that TCR tilt brings the two membranes in closer juxtaposition. But that would only be true if MHC would also be sufficiently flexible to compensate for the TCR tilt, keeping the two membranes parallel.

We agree – the assumption that the MHC complex is more flexibly anchored than the TCR EC domain withing the TCR – CD3 complex is a central assumption of our calculation and now clearly stated and discussed both in the Results and Methods section.